# The Role of Toll-like Receptor Agonists and Their Nanomedicines for Tumor Immunotherapy

**DOI:** 10.3390/pharmaceutics14061228

**Published:** 2022-06-10

**Authors:** Lingling Huang, Xiaoyan Ge, Yang Liu, Hui Li, Zhiyue Zhang

**Affiliations:** Department of Pharmaceutics, Key Laboratory of Chemical Biology (Ministry of Education), School of Pharmaceutical Sciences, Cheeloo College of Medicine, Shandong University, 44 Wenhuaxi Road, Jinan 250012, China; huanglingling@mail.sdu.edu.cn (L.H.); gexiaoyan@mail.sdu.edu.cn (X.G.); liuyang_sdu@163.com (Y.L.)

**Keywords:** toll-like receptors, TLR agonist, immunotherapy, nanomedicine, combined therapy

## Abstract

Toll-like receptors (TLRs) are a class of pattern recognition receptors that play a critical role in innate and adaptive immunity. Toll-like receptor agonists (TLRa) as vaccine adjuvant candidates have become one of the recent research hotspots in the cancer immunomodulatory field. Nevertheless, numerous current systemic deliveries of TLRa are inappropriate for clinical adoption due to their low efficiency and systemic adverse reactions. TLRa-loaded nanoparticles are capable of ameliorating the risk of immune-related toxicity and of strengthening tumor suppression and eradication. Herein, we first briefly depict the patterns of TLRa, followed by the mechanism of agonists at those targets. Second, we summarize the emerging applications of TLRa-loaded nanomedicines as state-of-the-art strategies to advance cancer immunotherapy. Additionally, we outline perspectives related to the development of nanomedicine-based TLRa combined with other therapeutic modalities for malignancies immunotherapy.

## 1. Introduction

Cancer is a leading cause of death worldwide and remains extremely demanding to cure [1]. To date, immunotherapy contains manifold therapies to engage the immune system in order to target malignancies and has had a principal influence on the treatment of metastatic cancer and has altered the standard of care for many tumor types [2]. Recent research on the adoption of immunomodulation to improve the effects of targeted cancer treatments has resulted in the development of immunoadjuvants [3]. Immunoadjuvants are molecules that tumors evolved as antigen sources to provoke tumor-specific immune responses. TLR agonists are one of the immune adjuvants that function by activating antigen-presenting cells and initiating immune responses, hence their study is sought after as potential vaccine adjuvants and cancer immunotherapies [4].

Toll-like receptors (TLRs), a family of pattern recognition receptors [5], can sense and initiate innate and adaptive immune responses [6] by responding to exogenous infectious ligands (pathogen-associated molecular patterns, PAMPs) and endogenous molecules released during host tissue damage/death (damage-associated molecular patterns, DAMPs) [7], such as lipopolysaccharide (LPS) and heat shock proteins (HSP) released from apoptotic or necrotic cell bodies [8]. As a class of complete type I membrane glycoproteins, TLRs are characterized by an extracellular recognition domain consisting of leucine-rich repeats (LRRs) for ligand binding, a single transmembrane domain, and an intracellular Toll/IL-1R homology (TIR) signaling domain [9,10,11]. Owing to the faculty specifically recognizing PAMPs through TLRs, the downstream signaling cascade is launched, subsequently initiating innate and adaptive immune cells, such as dendritic cells (DCs), macrophages, and T cells (Figure 1) [12]. Among a series of antigen-presenting cells (APCs), dendritic cells (DCs) are the most considerable anticancer immune cells owing to their ability to collect and process antigens to present immunogenic epitopes [12]. At the cellular level, this process constructs proinflammatory cytokines (TNF-α, IL-6 and IL-12), upregulates costimulatory molecules (CD40, CD80 and CD86), facilitates antigen presentation capacity, and migrates DCs from peripheral tissues to draining lymph nodes (LNs) [13]. However, STAT3 signaling, as a member of signal transduction and transcriptional activators (STAT) proteins family, has been shown to play a crucial role in DCs function. It is of note that the self-secreted tumor-derived factors (TDFs) IL-6, VEGF, and G-CSF can stimulate the STAT3 cascade in myeloid cell differentiation, impairing DCs generation and function in breast cancer TME [14]. Similarly, naive T cells receive a signal from the initiation of APCs, which induces T cells into a highly proliferative state and stimulates them to upregulate the expression of various effector/cytolytic molecules, including IFN-γ, perforin, and granzyme. The surviving T cells become long-lived memory T cells with the capacity for self-renewal and the ability to respond to antigen stimulation within hours of exposure. The involvement of specific TLR on effector T cells contributes to antitumor activity and T cell survival, and TLR signaling in memory T cells may assist in maintaining their homeostasis [15]. Based on their activation phenotype, macrophages with antitumor or cytotoxic activity are termed M1, while tumor-promoting or healing macrophages are named M2 or M2-like [16]. Nevertheless, the ERK/STAT3 cascade has become a key regulator of stimulating the M2-like polarization of macrophages and promoting tumor progression and metastasis in breast cancer TME [14]. Conversely, the cytokine interferon-γ (IFN-γ) is a type II IFN that induces anti-tumor M1 macrophages in collaboration with TLRa [17]. Furthermore, IFN-γ collaborated with TLR agonists for the induction of macrophage tumoricidal activity and the production of both NO and proinflammatory cytokines (TNF-α, IL-12p40, and IL-12p70), thereby boosting macrophage-based cancer immunotherapy [16].

Innate immunity plays a crucial role in initiating the immune response prior to activating adaptive immunity [18]. For example, toll-like receptor (TLR) innate immune genes can also be regulated by the transcription factor and oncosuppressor protein p53 to alter the immune system in response to DNA stress in cancer cells. In addition, p53 regulates endogenous antigen presentation through transcriptional control of aminopeptidase ERAP1 and peptide transporter TAP1. Both MHC classes have the task of presenting neoantigen peptides on the cell surface for T cells to recognize. For example, p53 targeting TLR3 and TLR9 activates their expression and initiates apoptosis [19]. As a consequence, TLR signaling acts as a vital regulator in the immune response against cancer and is of potential application in cancer immunotherapy [20].

Advanced nanomedicine design, including active-targeting, tumor-sensitive, and the optimization of physicochemical properties and regulation of the tumor microenvironment, remains the most mainstream treatment method adopted in the clinic to promote the delivery of nanomedicine to the tumor milieu [22]. With the rapid development of nanotechnology, the integration of nanodrugs that can progress therapeutic effectiveness and lessen immune-related side effects has become a topic of widespread discussion in the field of tumor immunotherapy [23].

In this review article, we first briefly depicted the patterns of TLRa, followed by the mechanism of agonists towards those targets. Second, we summarized the emerging applications of TLRa-loaded nanomedicines as state-of-the-art strategies to improve cancer immunotherapy. Finally, we outlined perspectives related to the development of nanomedicine-based TLRa combined with other therapeutic modalities for cancer immunotherapy (Figure 1).

## 2. TLRa in Tumor Therapy

Intensified efforts have increasingly been made to elicit immunity in CD8^+^ T cells, for instance, a structurally diverse class of synthetic and naturally occurring molecules that bind to specific TLRs [24]. TLRs, found on the cell surface, respond to the extracellular components of pathogens, including lipoproteins (TLR1, 2 and 6), lipopolysaccharides (TLR4), and bacterial flagellin (TLR5). Endosomal TLRs recognize components from the intracellular compartment of the pathogen, such as double-stranded RNA (dsRNA, TLR3), single-stranded RNA (ssRNA, TLR7 and 8), and unmethylated cytosine–phosphate–guanine (CpG) DNA (TLR9), while a specific ligand for TLR10 has not been established. In particular, it is noteworthy that TLR11 is not expressed in humans and can recognize urinary E. coli in mouse macrophages [13,25,26]. To date, 13 TLRs have been identified in mammals (TLR1-TLR13), 10 of which are encoded in the human genome (TLR1-TLR10) [9]. Notably, the human TLR11 is pseudo-genetic, and human cells are deficient in TLR12 and TLR13 [9].

The biology of TLR signaling has been reviewed thoroughly and in detail elsewhere [27,28]. Herein, we briefly and succinctly introduce the mechanism of TLRa (Figure 2). Four adaptors are currently known: MyD88, TIRAP (Mal), TRIF (TICAM1), and TRAM [29]. All TLRs, except TLR3, recruit signals through a common adapter molecule, MyD88, leading to the migration of NF-κB to the nucleus, the stimulus of the NF-κB transcriptional function and the expression of inflammatory cytokines and chemokines [10,20]. TLR3, however, activates NF-κB by the complexation of interferon-β (TRIF) with an adaptor containing the TIR domain [30]. Moreover, TLR4 can utilize both pathways [31] when the MyD88 TIRAP signal pathway results in the expression of proinflammatory cytokines. It is noted that TRAM-TRIF signaling is distinct from the MyD88 pathway and induces IFN-β and IFN-inducible gene expression [29].

Considering their merits of DCs maturation and antigen presentation promotion, versatile TLRa viewed as anti-cancer drugs and/or vaccines to induce a powerful immune response has considerable appeal. For instance, heat-killed Mycobacterium tuberculosis (HKMT) has been employed as a TLR2 agonist to promote DCs maturation. A study performed by Lemdani et al. demonstrated tumor suppression through cytotoxic T cell penetration upon intra-tumoral injection of MTI-gel, a thermal adhesive for the local delivery of HKMT and GM-CSF [32]. acGM-1.8, a glucomannan polysaccharide with acetyl modification, was another TLR2 agonist that triggered macrophages to express the desirable group of proinflammatory factors through activating TLR2. Our findings confirmed that acGM-1.8 administrated at 5 and 20 mg/kg had a 100 and 90% survival rate, respectively, remarkably safer than the four classical TLRa of LPS, Monophosphoryl lipid A (MPLA) (TLR4), Poly(I:C) (TLR3), and Pam3CSK4 (TLR1/2). It is well known that certain cytokines can be a double-edged sword, depending on their dose of expression. Therefore, it is necessary to control the duration and threshold of TLR activation more precisely [33]. BCG was originally developed as a tuberculosis vaccine more than 100 years ago, which was known as an attenuated strain of Mycobacterium bovis, and induced biologically active cytokines, such as IFN-γ and IL-2, and immune effector cells that improve tumor recognition and killing capacity through both nonspecific and specific (such as antigen-specific T cells) mechanisms [34,35]. Currently, it has been approved as a TLR2/4 mixed agonist for bladder cancer and superficial bladder cancer treatment. Furthermore, Poly(I:C)/BCG therapy could improve the therapeutic index for the BCG vaccine, which significantly inhibited the growth of MBT-2 tumors and induced immune memory response [36]. An endogenous TLR2/6 activation domain, human cysteinyl-tRNA synthetase 1 (CARS1), was the foremost and unique characteristic of high immunostimulatory activity as well as low toxicity. A unique domain (UNE-C1) inserted into the catalytic region of CARS1 was determined to motivate DCs, leading to the stimulation of robust humoral and cellular immune responses in vivo. Moreover, UNE-C1 can be employed as an extraordinary immune adjuvant in conjugation with checkpoint inhibitors or cancer antigens to enhance antitumor immunity [37]. Polyriboinosinic acid-polyribocytidylic acid (poly(I:C)) is a dsRNA analog and has recently been reported to inhibit tumors straightly by reducing proliferation and inducing apoptotic cell death in vitro [38,39]. Inspired by this mechanism, the nanocomposite formulation of poly(I:C) and polyethyleneimine, namely BO-112, is currently in phase I/II clinical trials (NCT02828098) [38]. Similarly, polyinosinic-polycytidylic acid polylysine carboxymethylcellulose (poly-ICLC)—a synthetic Poly(I:C) (TLR3 and MDA5 agonists)—can induce immune response mediated by tumor-specific NK, CTL, and NK-T cells, leading to the robust elicitation of multiple IFNs [40,41]. Furthermore, Poly-ICLC proved to increase initial antigen-specific CD8^+^ T cells and elongated survival in the glioma and melanoma cancer models [41].

MPLA, a non-toxic TLR4 ligand derived from LPS, has been approved as a Th-1 skewing adjuvant in cancer, hepatitis, and allergen-specific immunotherapy in the clinic [42]. Unlike other TLR agonists, TLR4 triggered by MPLA can simultaneously initiate MyD88 and TRIF pathways, leading to the driving force of DCs [43]. After this activation, antigen-presenting cells (APCs) secreted pro-inflammatory cytokines, including IFN-α, which further promoted the proliferation of CD4^+^ T cells and IFN-γ, TNF-α, perforin, and granulation produced by CD8^+^ cytotoxic T cells (CTLs) [44]. However, MPLA exhibited a qualitatively similar, but weaker, proinflammatory immune response performance compared with LPS [42], which was an endotoxin on the outer membrane of Gram-negative bacteria [44]. CRX-527, a conjugatable TLR4 ligand, is a potent robust lipid A analog for the generation of novel conjugated vaccine patterns. Incorporating the CRX-527 ligand into the N-terminus of the model peptide antigen provides a vaccine model that has been demonstrated to activate DCs, amplify antigen presentation, and initiate the specific CD8^+^ T-cell-mediated killing of antigen-loaded target cells in vivo [45]. Another synthetic TLR4 agonist, GLA-SE, upon single systematic administration, rapidly and dose-dependently increased innate and adaptive immunity in each compartment, with no apparent off-target effects, and resulted in a reduction in metastatic progression in all tumor models examined. More importantly, GLA-SE is considered a safe and well-tolerated adjuvant in phase II clinical trials in comparison with other agonists to date [44].

Imiquimod (R837) is a synthetic small molecule TLR7 agonist, a member of the imidazoquinoline family, and a nucleoside analog that specifically recognizes viral ssRNA to stimulate the secretion of type 1 cytokines and up-regulate immuno-costimulatory molecules [46,47]. A study performed by Oyama et al. demonstrated that the number of mast cells elevated with R837 in the treatment of actinic keratosis injury. The present result suggests that mast cells may suppress human skin cancer treated with topical R837 [48], which can induce apoptosis, cell cycle changes, and the up-regulation of myeloid differentiation markers in some cell lines [49]. Although R837 binds well to TLR7, it binds less effectively to TLR8, which led to the development of the TLR7/8 agonist MEDI9197 (3M-052). MEDI9197 is designed with a lipid tail to reduce aqueous solubility in order to exhibit high injection site accumulation and manipulate the tumor microenvironment to an inflamed immunophenotype [50]. To further reduce adverse reactions to system exposure, thermogel-based MEDI9197 upon intratumoral administrations has a significant reduction in drug diffusion and escape from the tumor site, resulting in sustained release [51]. Potent immune thrust was a consistent feature of the guanosine-and uridine-rich single-stranded RNA (Gu-rich RNA), another TLR7/8 agonist. Taking Gu-rich RNA a step further, Fusae Komura and colleagues designed a GU-RNA/DNA hydrogel (RDgel), where drugs could be continuously released [52]. Moreover, linking TLR7/8 agonists with checkpoint inhibitors and costimulatory agonists has also been shown to facilitate antitumor effects [50].

The human TLR8 ligand, expressed in myeloid DCs, monocytes, and monocyte-derived DCs, motivates unique cytokine profiles that assist in the development of Th1 [53]. Motolimod (VTX-2337) is known as a TLR8-selective small-molecule agonist that stimulates NK cells, DCs, and monocytes, which is being tested in a phase II clinical trial in patients with SCCHN and is currently used in combination with cetuximab with acceptable toxicity [54]. In addition, 5-(4-aminobutyl-N^4^-butyl-6-methyl pyrimidine-2,4-diamine, a newly synthesized TLR8 agonist, induces Th1-biased IFN-γ and IL-12 in human blood but minimizes the levels of proinflammatory cytokines IL-1β, IL-6, and IL-8 [55]. Because of its propensity towards inflammatory and reactogenic effects, the compound could be massively more profitable than other TLR8 agonists under evaluation. Furthermore, 17e (CU-CPT17e) is another innovative TLR3/8/9 agonist that produces many-sided cytokines in human monocyte THP-1 cells and hinders the proliferation of HeLa cancer cells by inducing apoptosis and blocking the cell cycle in the S phase [56].

The TLR9 agonist, CpG-oligodeoxynucleotides (CpG-ODN), a synthetic bacterial sequence derived nucleotides with a phosphorothioate backbone chain and repeated CpG motif [57], serves as a promising target for innate immune response in the bone marrow and the almost complete elimination of B-ALL cells [58]. Meanwhile, CpG-ODN (GNKG168) was discovered to be involved in checkpoint signal suppression and upgrading in T cells, B cells, and innate immune responses in pediatric leukemia patients [59]. However, CpG comes at the expense of limited efficiency in inducing T cell immune responses clinically, possibly because TLR9 is expressed merely in human B cells and plasmacytoid dendritic cells (pDCs) [23]. In addition, CpG ODNs are associated with the upregulation of indoleamine 2,3-dioxygenase (IDO) side-effects, and the enzymatic properties of IDO have been reported to inhibit T cell activation and promote Treg induction to the tumor milieu [3]. This adverse reaction would be interesting to explore further, given that Melssen and colleagues noted that lower levels of indoleamine 2,3-dioxygenase 1 (IDO1), as well as CD8 inhibitory pathways, programmed death-ligand 1 (PD-L1), were committed to a cancer vaccine containing CpG ODN in a liposomal formulation [60].

## 3. Combination of Multiple TLRa for Tumor Therapy

In the context of multiplexed applications of TLRa in preclinical and clinical trials, several newly synthetic TLRa have been further optimized based on classical agonists to improve pharmacokinetic and stability in vitro. Nevertheless, single TLRa with deficient immune response hinders widespread adoption. Thus, the combination of multiple TLRa has been properly and adequately studied and will be discussed in detail as follows.

Therapies based on multiple TLR incentives have considerable merits in clinical application over a single one and the reasons for this can be summarized as follows: First, a key aspect of versatile TLR activation can exhibit synergies between multifunctional TLR pathways concurrently in separate cell compartments while invading pathogen release components. Second, when various invading microorganisms are incorporated in one site of infection, the invading organism is more likely to induce simultaneous activation of several TLRs, sometimes even involving other pattern recognition receptors, which enables the host to initiate a powerful immune response to eliminate the invading pathogens. Additionally, TLRs are expressed differently in tissues and cell types, and even within the same cell type, TLR expression patterns differ in diverse subpopulations significantly. The involvement of considerable TLR replenishment to stimulate DCs simultaneously achieves a pivotal activation threshold for a robust immune response [56,61]. Eventually, synergistic multivalent TLRs display superior antitumor efficacy while reducing systemic off-target toxicity [62], which pays the way for tumor immunotherapy. Synergistic immune activation with two chemical molecules: Poly(I:C)—a TLR3 targeted agent—and TLR2/4 inducer-BCG led to tumor suppression and the generation of a long-lasting protective immunity in the MBT-2 murine high-grade bladder cancer model [36]. Moreover, BCG combined with R837 can increase BCG potential tremendously by up-regulating TLR7/4 and down-regulating the P70S6K1 protein, which are able to reduce the incidence of bladder cancer powerfully by controlling tumor proliferation and apoptosis [35]. Peptidoglycan (PGN) and Poly(I:C), which are TLR2 and TLR3 agonists, respectively, enhanced the T-cell initiation ability of skin-migrated DCs and were accompanied by Th1 polarization in the case of PGN. Increased activation of poly(I:C)-stimulated DCs was associated with the strong growth of appropriate costimulatory molecules, including CD70, while PGN-stimulated DCs were involved in the release of versatile proinflammatory cytokines [63].

Triggering or enhancing the antitumor activity of tumor-associated macrophages is an attractive cancer treatment strategy. For example, the binding of two TLR ligands, Poly(I:C) and Pam3CSK4, activated murine bone-marrow-derived macrophages (BMDMs) effectively with an anti-tumor phenotype through Poly(I:C)-induced autocrine IFN-I signaling. Notably, Poly(I:C) embedded in nanoparticles can enhance the synergistic effect with Pam3CSK4 and induce antitumor macrophages up to 100 times [17]. However, bacterial LPS, a TLR4 agonist, inhibited the production of TLR3-induced proinflammatory cytokines in DCs, which were mediated by IL-10. The ability of LPS to mediate IL-10-dependent inhibition of Poly(I:C)-induced inflammatory response suggests a sophisticated interaction between TLRa [64].

Furthermore, in a series of TLRa tests, the data showed that TLR8 with either TLR4 or TLR3 programed DCs to produce CD4 (Th1) and CD8 (Tc1) effector T cells. Interestingly, TLR4+ 8 combination showed superior adjuvanticity to polarize the DC-NK mediated differentiation of naïve T cells into both effector memory CD4, especially CD8 cells producing IFN-γ [65].

Therefore, certain TLRa combinations are synergistic while others have an antagonistic effect. It is interesting to note that the selection of adjuvants and their combinations, considering DCs and NK cells crosstalk and the magnitude of desired cellular responses, receive significant attention.

The combination injection of Poly(I:C) with resiquimod/R848 in humanized mouse models (hu mice) further enhanced the expression of CD141^+^ and CD1c^+^ DC co-stimulatory CD80, CD83, and CD86 [66]. Furthermore, the combination of Poly(I:C) and R848 with E7 DNA can induce the enormous regression of tumors, circulating antigen-specific IFN-γ and non-specific intratumoral IL-12 [67]. Similarly, Toll-like receptor agonists (TLR-P), Poly(I:C)/R848/PGE2, contribute to DCs with mature phenotypes and a brilliant ability to migrate and secrete cytokines, especially IL-12p70 production [68]. In the cancer therapy realm, work by Caisova and coworkers optimized therapeutic mixtures based on well-defined compounds, including the biocompatible anchor for the membranes (BAM) of mannan anchored to the surface of tumor cells, R848, poly(I:C), and lipoteichoic acid (LTA), leading to the eradication of advanced stage progressive melanoma in 83% of mice, the acquisition of resistance to tumor recurrence, and potential anti-metastatic effect [69]. Overall, it has long been known that TLR agonists can exhibit the synergistic activation of DCs; for example, R848 combined with Poly(I:C) and other agents can synergistically stimulate inflammatory cytokines in versatile human DC subpopulations.

To further develop the idea that robust immune responses in vivo from simultaneous activation of TLR9 and other agonists can be clinically translated, it has been shown that endogenous (i.e., autologous) non-toxic TLR4 agonist additional domain type A III repeat fibronectin (FNIII EDA) works synergistically with exogenous (i.e., bacterial) highly toxic TLR9 agonist CpG, resulting in the increased activation of DCs and CTLs as well as more intensive humoral responses, even at half doses, compared to the individual agonists given at maximum doses [70]. Similarly, the combined delivery agonists 3M-052 and CpG ODN targeting TLR7/8 and 9 increased CTLs and NK cell activity while down-regulated the faculty of immunosuppressive myeloid-derived suppressor cells (MDSCs), thereby eradicating bulky primary tumors and establishing long-term immune protection [71]. The TLR3 agonist, Poly(I:C), and the TLR9 agonist, CpG ODN 1826 [CpG], upon intra-tumoral administration, combined with the systematic metastasis of specific gp10025-33-activated pmel-1 T cells, enhanced IFN-γ production and the immunogenicity of B16F10 melanoma cell lines and improved the survival rate and eradication of subcutaneous 9-day B16F10 melanoma in some mice [72]. It is worth noting that when combined with poly(I:C) and CpG ODN for cancer treatment, these drugs should be adopted alternately, rather than together, to avoid the blocking effect of thiophosphate-modified TLR9 ligands and enhance proapoptotic effects [73]. These optimistic data may pave the path for the development of adoptive T-cell cancer immunotherapy with the hope of the clinical enhancement of this approach or in combination with other therapies.

In light of the current studies of TLRa, the combined use of TLRa would be expected to shed light on the single TLRa for augmenting the production of manifold types of cytokines, activating tumoricidal cells potently, reducing the frequency of immunosuppressive cells, and inhibiting tumors growth. Moreover, combination therapy also curtails drug resistance and has a definite inhibitory effect on tumor regeneration and metastasis partly. What is more, selecting the right combinations is foremost [74], since combinations of TLRa are either synergistic or inhibitory, altering cellular and antibody responses [75]. As a consequence, it is important to better define the crosstalk between different TLRs so that appropriate combinations of TLR ligands may be developed in antitumor therapies [73].

## 4. Application of Single TLRa Nanoparticles in Tumor Therapy

We next highlight and discuss recently reported strategies exploring TLR agonists as adjuvants in nanomedicine against cancer (Table 1). The positive sides of TLRa are apparent in their practicality, yet they still exist manifold shortcomings. The adverse effects of TLRa are closely linked to their adverse PK/PD status [76]. For instance, intra-tumoral injections of TLRa, such as CpG (TLR9 agonist) and imidazoquinoline (TLR7/8 agonist), are distributed systematically, driving the systemic proinflammatory cascade and severe immune-related toxicity. Importantly, TLRa-mediated immune responses are only selectively initiated in ideal environments to avoid systemic side effects [77]. As a consequence, an approach that restricts immune activation to a specific site would improve the treatment window exceptionallly. In the meantime, nanomedicines have attracted increasing attention on account of their superior tumor targeting efficiency and biocompatibility [1]. In this review, we have studied various materials, such as organic particles, like PLGA [78,79,80,81] and liposomes [82], metal-organic frameworks (nMOFs) [83], or inorganic particles that resemble Iron Oxide NPs (IONPs) [84]. While the organic materials might be beneficial in biocompatibility or biodegradation, the inorganic materials can be valuable as theranostic vectors. In order to avoid cytokine release storms, the local and sustained release of cytokines in the body can be achieved through controlled-release techniques. Overall, the immunomodulatory properties of nanoparticles are associated with their particular range of particle size, many nanostructures have been reported to stimulate the immune response inherently through different TLR-mediated pathways [10]. Hence, most researchers are working on nanomedicine strategies that alter the pharmacokinetics of conjugated drug molecules [76].

### 4.1. TLR3a

Poly(I:C) is a synthetic dsRNA mimic that activates immune cells via TLR3 [100]. BO-112 is a nano-compound preparation of poly(I:C) and polyethylenimine (PEI) with the characteristics of tumor cell apoptosis induction and immunogenic cell death [38]. Upon intratumoral injection, BO-112 results in more prominent CTLs infiltration, leading to extraordinary local disease control, dependent on IFN-I and IFN-γ in MC38, 4T1 and B16-F10, three subcutaneous tumors. Nevertheless, in the case of BO-112, the expression of CD137 and PD-1/PD-L1 is increased by tumor-infiltrating T and NK cells, which requires the further investigation of the best combination therapy. Hitherto, BO-112 has been generally clinically employed in combination with anti-PD-1 monoclonal antibodies.

### 4.2. TLR4a

A myriad of research groups have investigated the LPS in Gram-negative bacteria that bind to TLR4 in order to create a cancer vaccine [101]. Despite their great potential in vaccines, TLR4 agonists have been confined in their efficacy as monotherapies against cancer [84]. To solve this problem, Traini and colleagues employed low lipopolysaccharides (LOS), LPS without O antigen chains (known as crude LPS), model antigens ovalbumin (OVA) linked to mIONPsp via hydrazone bonds, and adjuvants (Xcc LOS) linked to mIONPsp via hydrophobic interactions to generate nanostructures that mimic pathogens (Figure 3A–C). The nanovaccine system yielded higher levels of antigen-specific-CTL effects and memory response and provided 100% long-term protection against repeated tumor invasion when interfered with by immunosuppressive PD-L1 [84]. However, LPS has another shortcoming—strong proinflammatory properties, leading to the low tolerance of its local and systemic side effects, which restricted the dose administered. It is well known that PLGA is a prevalent candidate that loads numerous substances and then releases them rationally as designed [102]. Very recently, Shetab Boushehri et al. developed LPS-modified PLGA-based nanoparticles (LPS-NP) to simulate pathogens. Unlike local necrosis caused by high concentrations of LPS solution, tumor-bearing animals treated with an equivalent dose of LPS-NP were well-tolerated. Hence, nanoparticles can not only serve as carriers for surface-modified LPS molecules but also overreach their overall efficiency and tolerability in cancer therapy [78]. These results suggested the promising potential of TLR4 agonist-based NPs in the future development of cancer therapy.

### 4.3. TLR7a

Diversified TLR7 agonists nanostructures are conducted and comprehensively discussed below. TLR7, a member of the TLR family, is an intracellular receptor expressed on the endosomal membrane and can be triggered not only by ssRNA during viral infection but also by immune modifiers with a similar structure to nucleosides. The mechanism of TLR7 agonists is related to MyD88 dependent pathway and the Caspase-dependent mitochondrial pathway [103]. Different types of nanocarriers, such as polymer- and lipid-based nanoparticles, metal nanoparticles, and inorganic nanoparticles, have been manufactured for the delivery of immune adjuvants [8]. Ni, K., et al. demonstrated that nanoscale metal−organic frameworks repolarized immunosuppressive M2 macrophages into immunostimulating M1 macrophages and modulated the immunosuppressive tumor microenvironment remarkably, leading to the activation of innate immunity [83].

The extraordinary possibilities of appropriate potential immunoadjuvant carriers are due to polymeric nanoparticles. To address the problem of immunosuppression induced by immature plasmacytoid dendritic cells (pDCs) in tumor sites, Shen et al. developed hypoxia-sensitive imiquimod (hs-IMQ) to activate IMQ selectively under the catalysis of poly(L-glutamate)-grafted-methoxy PEG/Comprestin A4 (CA4-NPs)-induced nitroreductase (NTR). The combination of hs-IMQ and CA4-NPs exhibited superior active IMQ concentrations by 6.3 times as compared with hs-IMQ alone. Additionally, the tumor microenvironment was transformed from an immunosuppressive state to an immuno-motivated state, leading to the drastic infiltration and activation of NK cells and CTLs and inhibited tumor growth and metastasis synergistically in mice bearing the 4T1 tumor [85]. In another study, Xu et al. obtained multifunctional UCNP-Ce6-R837 nanoparticles by employing up-transformed nanoparticles (UCNPs), photosensitizer chlorin e6 (Ce6), and TLR7 agonist R837. This work presented UCNP-Ce6-R837, an immune-stimulating UCNP-based PDT strategy, in combination with the cytotoxic T lymphocyte-associated protein 4 (CTLA-4) checkpoint blockade, to destroy primary tumors potently under light exposure, inhibit distant tumors that can hardly be reached by light, and prevent tumor reoccurrence via the immune memory effect [86].

Both TAMs and tumor-infiltrating dendritic cells (TIDCs) are crucial elements in the tumor microenvironment that manipulate tumor immunosuppression and promote cancer progression [88]. Huang et al. created a nucleic acid delivery system consisting of cBSP, PHA, and let-7b (TLR7 ligand and IL-10 inhibitor), which had an affinity for mannose receptors on TAMs /TIDCs and responded to low pH tumor microenvironments. As expected, they found that this system reprogrammed the function of TAMs/TIDCs effectively, reversed the immunosuppression microenvironment, and controlled tumor growth in a mouse model of breast tumors. In another approach, a newly developed TLR7 agonist TMX-202 prepared in liposomes also broadly enhanced mDCs and pDCs and upregulated the secretion of proinflammatory and anti-tumor cytokines [82]. Integrating this co-targeting delivery system with the dual regulating abilities of TLRa to reprogram DCs may represent a new device for cancer immunotherapy.

### 4.4. TLR7/8a

TLR7 is expressed by plasmacytoid DCs (pDC), which is a severe type I IFN inducer, and TLR8 by myeloid DCs (mDCs), which regulate costimulatory molecule-mediated signaling and proinflammatory cytokines [104]. The transcription of TLR7 and TLR8 is mediated by NF-κB in response to induced proinflammatory cytokine signals [105]. Therefore, mixed TLR7/8 agonists stimulate plasmacytoid DCs (pDCs) and myeloid DCs (mDCs) [104], promoting fertile immunomodulatory cytokines, such as interleukin-6 (IL-6), IL-12, and IFNα, triggering a series of signaling pathways that result in the activation of antigen-presenting cells (APCs) and the polarization of T cell responses [106]. However, the poor solubility of small TLR ligands hinders their systemic delivery to distal tumors and metastatic sites. Therefore, a practical and valid delivery system is urgently needed [89]. The development of anticancer therapies has proven to be challenging due to the rapid depletion of white blood cells and transient local immune deficiency with the systemic administration of R848 [106]. For these reasons, immunoadjuvant-loaded nanoparticles have been designed widely to increase antigen-specificity and promote their applicability. R848 is an antiviral imidazoquinoline derivative that activates immune cells via MyD88-dependent TLR7 and/or TLR8 [107]. It has been conjugated to α-tocopherol to form a R848-Toco prodrug, which was then combined with tocopherol-modified hyaluronic acid (HA-Toco) as polymeric nano-suspension. Upon subcutaneous injection, nano-suspension formed a reservoir at the injection site, inducing a local immune response without systemic dilatation and suppressing tumor growth in a murine model of head and neck cancer [94]. More recently, the poly (2-oxazoline)-based nanomicellar formulation of R848, POx, was an amphiphilic triblock copolymer, which stimulated DCs and macrophages and led to the mobilization of CD8^+^ T cells in vivo. Moreover, it exhibited a superior tumor inhibitory effect in a metastatic model of lung adenocarcinoma, as compared with anti-PD-1 therapy or platinum-based chemotherapy [89]. This concept will be interesting to explore further, given that Nuhn and his colleagues note that in localized treatment with IMDQnano, based on core-crosslinked block copolymers and combined with anti-PD-L1 checkpoint inhibition and Flt3L, a growth factor did show improved anti-tumor efficacy (Figure 3D) [76]. Furthermore, Chen et al. proposed an immune modulator (R848)-loaded nanoparticle system (R848@NPs), which was composed of an amphiphilic copolymer, a hydrophobic polyaniline (PANI) covalently bound to hydrophilic glycol-chitosan (GCS) chain, and R848 was contained in the hydrophobic core (Figure 4A). Following intra-tumoral injection, R848@NPs could absorb near-infrared light (+NIR) to cause low-temperature hypothermia that interacted synergistically with loaded R848 to alleviate the tumor-mediated immunosuppressive microenvironment, producing robust anti-tumor memory immunity and preventing cancer recurrence and metastasis [93]. Taking TLR agonists and STING agonists a step further, Seth and colleagues added a TLR7/8 agonist-Gardiquimod into PLGA nanoparticles combined with vasculature disrupting agent (VDA)—DMXAA—and observed a synergistic effect of immune stimulation caused by DCs activation and vasculature disruption [81]. Our therapeutic strategies may be confined to localized solid tumors, and immunotherapy as a pre-operative neoadjuvant therapy may be a feasible modality to reduce systemic toxicity and improve efficacy.

Apart from copolymer nanoparticles as carriers, we also studied the preparation of cyclodextrin TLR7/8 nanoparticles. More importantly, cyclodextrins formed water-soluble inclusion complexes with various insoluble drugs, enabling agents to be solubilized by hydrophilic modification of cyclodextrins and providing affinity for drug delivery when prepared in nanoparticles, surface coatings, or bulk materials. R848-loaded β-cyclodextrin nanoparticles (CDNP-R848) can be delivered to TAMs and with improved re-education of macrophages within the tumor microenvironment [91]. Furthermore, Rodell and colleagues designed R848-Ad, an adamantane modified derivative of resiquimod. In addition to polarizing macrophages, adamantane enhanced the affinity for cyclodextrin nanoparticles (CDNPs) partially [92]. This approach was a promising strategy to reposition the TIME function within an immune-activated state by targeting TAMs. Moreover, the key interaction between nanoparticles and agents can hinder the systemic toxicity of TLR agonists while maintaining therapeutic efficacy.

In addition to organic polymerized nanoparticles, versatile membranes-based biomimetic nanosystems have also been developed, enabling TLR7/8 agonists to enhance biocompatibility and natural targeting affinity. Cell membranes-based biomimetic nanosystems show great superiority in biocompatibility, long blood circulation, and drug delivery [1]. Bahmani et al. designed a local delivery system of the TLRa R848, via platelet membrane-coated nanoparticles (PNP-R848) [90]. Following intra-tumoral injection, PNP-R848 enhanced the local immune activation massively and led to complete tumor regression, while protecting against recurrent tumors in the colorectal cancer models. In addition, among nanocomplex tested, a dramatic reduction in metastasis was achieved in an aggressive breast cancer model.

An imidazoquinoline-based small molecule (called “522”) evolved as a TLR7/8 mixture agonist and a more potent proinflammatory cytokine inducer compared to the canonical TLR7 agonist imiquimod [104]. The TLRa “522”, encapsulated in PLGA nanoparticles migrated the draining of LNs and triggered DCs activation and dilation, inducing the amplification of antigen-specific CD8^+^ T cells and the enhancement of the CTLs response [80], yet the loading rates of this conventional PLGA nanoparticles were confined. Stimulus-responsive nanomaterials are ubiquitous in the field of drug delivery and attempts to enhance the efficacy of the TLR7/8 agonist “522” employing this paradigm have made some progress in the following experiments. The same group optimized the original system, and subsequently the TLR7/8 agonist “522” was released specifically by an acidic pH-responsive PLGA nanoparticle. The new formulation incorporated bicarbonate salt, which generated carbon dioxide (CO_2_) gas at acidic pH that could break the polymer shell to release the payload rapidly. Notably, this system induced a stronger antigen-specific CD8^+^ T cell and NK cell response than conventional PLGA nanoparticles and the anti-cancer effect of the mouse melanoma model in vivo was further promoted [79]. Recently, Tumor vaccines remain the most common mainstream treatment method adopted in the cancer immunotherapy field. Yet, cancer vaccines have so far shown suboptimal efficacy in the clinic. Nanomedicines represent a unique chance to improve the efficacy of these vaccines [110]. Kim and colleagues studied a nanoparticle-based cancer vaccine to deliver TLRa “522” to target DCs, resulting in the reduction of both MDSCs and regulatory T cells (Tregs) [87]. Of note is the further binding of the thiolated nanovaccine with αPD-1 which could enhance tumor elimination and the complete prevention of tumor recurrence with a long-term survival of more than 150 days [111]. Therefore, creating an innovative cancer vaccine delivery strategy based on a nanosystem would be extremely attractive for practical merits and have good therapeutic potential to augment its cancer immunotherapy efficiency.

### 4.5. TLR9a

CpG ODNs, which bind the TLR9 ligand, are capable of promoting the proliferation of cytotoxic CD8^+^ T cells toward tumor antigens and inducing antigen-specific adaptive immune responses [112]. Unfortunately, nude CpG ODNs fail to penetrate the cell membrane and are cleared easily by nucleases, which may induce an inflammatory response in serum through systemic administration, hence applications are greatly restricted [96]. TLRa is delivered to target cells via nanoparticles in order to prolong circulation in the bloodstream and enhance agent accumulation in tumor sites. Moreover, nanomedicines could enlarge the therapeutic window and lessen immune-related side effects caused by drug distribution in normal tissues [113]. For instance, PEI-CpG nanocomplex could enhance the stability and cellular internalization of CpG. Moreover, nanocomplex increased the infiltration of NK cells and T cells in tumors prominently as well as the expression of CD80 on DCs, which inhibited the growth of murine B16F10 melanoma [98].

Taking nanomedicine a step further, being made to target TAMs and repolarize the proinflammatory M1 type in order to kill the tumor, Shan et al. designed a nanoparticle vector for the targeted delivery of CpG ODNs to M2-type TAMs by encapsulating CpG ODNs inside human ferritin heavy chain (rHF) nanocages surface modified with murine M2 macrophage-targeting peptides, M2pep. Upon intravenous administration, these M2pep-rHF-CpG nanoparticles repolarized M2-TAMs into M1-TAMs and inhibited tumor growth in mice bearing 4T1 tumors [96]. In another attempt, cholesteryl pullulan (CHP) nanogel following intravenous injection was able to deliver a long peptide antigen to TAMs and, when accompanied by CpG oligoDNA, TAM-targeted antigen delivery was able to broadly elicit antigen presentation by TAMs, thereby transforming the resistant tumors into being sensitive to adaptive immune responses [97]. However, nanoparticles face the problem of low uptake by APCs and defective retention in LNs, which undoubtedly limits their ability to activate T cell immunity. To address this issue, Zeng et al. fine-tuned the physical and chemical properties of polymer hybrid micelles (HMs)-two amphiphilic diblock copolymers, onto which, melanoma antigen peptides Trp2 and TLR9 agonist CpG ODN were encapsulated into sub30 nm HMs, which could expand antigen-specific CTLs drastically and provide a powerful anti-tumor effect in a lung metastatic melanoma model [99]. To conclude, a large number of trials of TLRa nanomedicine cooperate with the progressed stabilization of TLRa and improved APCs activation, repolarized TAMs, facilitated the ablation of tumors and inhibited metastasis and recurrence.

## 5. Application of Multiple TLRa Nanoparticles in Tumor Therapy

Encouraging results are emerging from systems that exploit TLR signaling, nanotechnology, checkpoint inhibition, and molecular imaging for cancer immunotherapy [114]. The combination of TLR ligands potentiates immune response by providing synergistic immune activity by triggering distinct signaling pathways and may impact antigen-dependent T-cell immune memory. Nonetheless, their rapid circulation time on account of nuclease attacks hampers their clinical performance [115]. Accordingly, accumulating efforts in Table 2 have focused on the application of versatile TLRa combined with nanoparticles in tumor immunotherapy. Overall, the results demonstrate a rational design of adopting several TLRa in a context-dependent manner for efficient nanoparticulate adjuvant-vaccine development [116].

### 5.1. TLR3a + TLR7a

The development of immunomodulatory drug-loaded NPs targeting immune cells renders the virtue that even small amounts of immunomodulatory-loaded drugs may be abundant in order to achieve robust antitumor efficacy. For instance, magnetic nanoparticles may display original mechanisms to control the release, cell fate, and biological distribution of the drugs, adopting external magnetic stimulation operations, such as field-guided localization and hyperthermia, to enhance drug administration and/or combination therapy effects. Many magnetic-sensitive systems have been reported, including one developed by Bocanegra Gondan and colleagues, which is based on phospholipid micelles loaded with zinc-doped iron oxide magnetic nanoparticles conjugated by poly(I:C), R837, and the OVA antigen, triggering a robust immune response [114]. Furthermore, the nanovaccine decreased immunosuppressive PD-L1 performance when co-administered with immune checkpoint abrogation. These results and the simplicity and versatility of the system provided a good framework for future clinical transformation.

### 5.2. TLR3a + TLR9a

TLR3 and TLR9 are both located in endosomes, so immunostimulants must be endocytosed in order to reach their targets. While Poly(I:C) and CpG both have promising applications in cancer immunotherapy, a major challenge is to determine how to correctly deliver these potential treatments to tumor tissues. Highly positively charged polycation, poly-l-lysine (PLL), is widely utilized in intracellular genetic material (DNA or RNA) transmission research. PLL + CpG polymers are an excellent candidate for increasing intracellular transport and reducing transport away from the tumor [39]. Another recent attempt at intraperitoneal administration of Poly(I:C) and CpG ODN was reported by Bayyurt and coworkers, they developed a liposomal carrier system to co-encapsulate two agonists along with protein antigens as immune adjuvants, which improved the activation of DCs and the germicidal ability of mature macrophages, as well as the ligand internalization ability. Furthermore, this system produced a persistent anti-cancer immune response, amplified Th1 immunity and upgraded OVA-specific memory CD8^+^ T cell response, so as to be developed as a preventive cancer vaccine [115].

### 5.3. TLR4a + TLR9a

Specifically, Monophosphoryl lipid A (MPLA, TLR4 agonist) and CpGrich oligonucleotides (CpG, TLR9 agonist) stimulate dendritic cells (DCs) through two different pathways. MPLA induces IFN-β (TRIF) through the Toll/IL-1R domain adapter, triggering the interferon regulatory factor 3 (IRF3) pathway, while CpG induces the major myeloid differentiation response gene 88 (MyD88) [9,28]. One approach to further promote the activation of DCs were synthetic high-density lipoprotein nanodiscs co-loaded with MPLA and CpG and mixed with protein antigens. Kuai et al. demonstrated this method generated a powerful humoral immune response, including the induction of IgG responses. Moreover, the combination of ND-MPLA/CpG with OVA increased 8-fold antigen-specific CD8^+^ T cell response pivotally, promoting the regression of B16F10-OVA melanoma compared with free adjuvant [120]. To more directly target DCs, Zhu and colleagues developed a mesoporous silicon vector (MSV), composed of a melanoma-derived tyrosinase-associated protein 2 (Trp2) peptide and dual TLRa targeting the same DCs. They observed that the system co-delivered CpG oligonucleotide and MPLA together with Trp2 peptide, orchestrating robust host immune responses, including CD8^+^ T cells, CD4^+^ T cells, and macrophages, extending median survival significantly in tumor-bearing mice [119]. The mode of action of α-Galcer is largely due to its strong binding to differentiated clusters (CD1d), members of the CD1 family that express glycoproteins on the plasma membrane of antigen-presenting cells (APCs), such as dendritic cells (DCs). When the α-Galcer-CD1d complex is presented to the semi-invariant T cell receptor (TCR) on immutable natural killer T cells (iNKTs), a ternary complex is formed, leading to the secretion of various cytokines [122]. More recently, Sainz et al. combined the NKT agonist α-GalCer, melanoma-associated peptide antigens as well as TLR ligands CpG and MPLA with nanoparticles, which promoted DC maturation, consequently leading to the up-regulation of co-stimulatory markers, such as CD80/CD86, and the production of IL-12/CCL17. The chemokine CCL17 attracted CCR4^+^ CD8^+^ T cells, boosting the DCs’ ability to attract effector cells [118]. These strategies aiming at antigen-presenting DCs may play a key role in solid tumor therapy into practical reality.

### 5.4. TLR4a + TLR7/8a

In a similar vein, in order to enhance DC targeting and vaccine efficiency, Zhang et al. designed a lipo-polymer hybrid nanoparticle delivery system that was attached to TLR7/8 agonist (R837), TLR4 agonist (MPLA), and the model antigen. Immunization with MAN-OVA-IMNPs induced antigen-specific CD8^+^ T cells, greater lymphocyte activation, stronger cross-presentation, more generation of memory T cells, and more (Figure 5A), and when combined with the immune checkpoint blockade, further enhanced the anti-tumor effect [43]. The same group fabricated mannose-functionalized lipid-hybrid polymersomes (MAN-IMO-PS) for the co-delivery of the OVA antigen both inside the inner core and outside the lipid layer; R837, within the hydrophobic membrane, MPLA in the lipid layer (Figure 5B). MAN-IMO-PS was internalized by DCs profitably via mannose targeting and TLR4 ligating, stimulating greater lymphocyte activation, CD4^+^ and CD8^+^ T cell responses, effector cytokine secretion, and inducing Th1 biased humoral responses [117]. This study demonstrated a properly designed nanovaccine that could bind antigens to different TLR agonists and target portions in a programmed manner to induce a synergistic antitumor immune response.

### 5.5. TLR7/8a + TLR9a

In non-human primates, the combination of TLR7/8 and TLR9 agonists upgrades the induction of human immunodeficiency virus envelope glycoprotein neutralizing antibody titers. When immunized with PLGA, NPs coated with TLR4 and TLR7/8 agonists, in combination with two soluble recombinant antigens of simian immunodeficiency virus, induced strong innate and antigen-specific antibody immune responses [116]. However, the co-use of adjuvants and antigens may not guarantee the stability of the immune response, as the soluble formulation of the vaccine can lead to confusion in vivo, limiting the immunogenicity of the vaccine against the tumor [121]. Ebrahimian and colleagues designed PLGA/PEI NPs, which were encapsulated with R848 or MPLA (inside the NPs) and CpG ODN (outside the NPs) and co-delivered with OVA [116]. The resultant robust cytokine (IFN-γ, IL-4, and IL-1β) secretion and antibody (IgG1, IgG2a) production reinforced dual TLR agonists in a context-dependent manner for highly potent nanoparticulate adjuvant-vaccine development. Likewise, a bi-adjuvant neoantigen nanovaccine (banNV) was developed for potent cancer immunotherapy by co-delivering a peptide neoantigen (Adpgk) with two adjuvants (TLR 7/8 agonist R848 and TLR9 agonist CpG) (Figure 5C). BanNVs could co-deliver adjuvants and neoantigen to LNs and LN-residing APCs, mediating the efficient uptake and presentation of antigens on APCs and augmented the neoantigen-specific cytotoxic T cell responses. Moreover, combined with αPD-1, BanNVs caused the complete regression of 70% of neoantigen-specific tumors without conspicuous recurrence [121]. In conclusion, this study suggests that banNVs have great potential in promoting the immunogenicity of tumor neoantigens and have a capacity for personalized tumor combination immunotherapy. The use of these combinations of such immuno-stimulatory or immune-modulatory adjuvants have revealed significant efficacy compared to their singular adoption, demonstrating that seeking optimal combinations of the currently available or well-defined adjuvants as well as antigens may provide a superior prospect for the development of adjuvants-vaccine in cancer immunotherapy [123].

## 6. TLRa-Based Combination Therapy

### 6.1. Combination of Immunotherapy and Chemotherapy

In the last section, we will discuss the current combination therapy design against cancer involving the TLR pathway and chemotherapy, radiotherapy, photothermal therapy, and immune checkpoint inhibitors and share our visions aimed at unresolved issues and open questions.

The combination of chemotherapy with immunoadjuvant therapies is viewed as a means of promising patterns in the treatment of manifold cancers. Besides killing cancer cells directly, some chemotherapeutic agents maintain the capability to alter the tumor microenvironment and strengthen the immune system [8]. Chemotherapy agents are dependent on DCs, such as doxorubicin (DOX), epirubicin, mitoxantrone, oxaliplatin [124], and cyclophosphamide (CTX) [125], which specialize in taking in tumor antigens and activating antigen-specific T cells of distinct subsets [126], triggering immunogenic cell death (ICD) and promoting anti-tumor immunity [124]. Low-dose rhythmic cyclophosphamide therapy has been shown to restore the activity of natural killer cells and T cells. However, CTX therapy alone can merely induce a confined anti-tumor immune response. Therefore, it is necessary to further explore cyclophosphamide combined immunotherapy therapy strategies. CTX chemotherapy can rapidly create a host microenvironment in bone marrow rich in proliferative DC precursors that can differentiate into functional DCs [127]. A combination of cyclophosphamide with either the TLR7/8 agonist (3M-011) or TLR7 agonists (852A) verified that cyclophosphamide did not negatively interfere with TLR agonists suppressing cancer but may actually potentiate the effect, depending on the dosing schedule [128]. Similarly, CpG (ODN 1826) or CpG+ Poly (I:C) in combination with CTX produced a unique and well-tolerated therapeutic synergy that permanently eradicated advanced mouse tumors, including 4T1 (breast), Panc02 (pancreas), and CT26 (colorectal). Tumor-specific IFNγ-producing T cells persist during cell-induced leukopenia, while Tregs are gradually eliminated, especially within tumors [129]. Some promising attempts have been made to design a temperature-sensitive PLEL hydrogel as a combined immunotherapy strategy. First of all, CTX-loaded hydrogel was intratumorally injected into CT26 mice to primarily spur anti-tumor immunity, three days later, PLEL hydrogel loaded with CpG and tumor lysates were subcutaneously injected into both groins of mice. This joint strategy represents another promising step in reducing the toxicity of CTX and producing a cytotoxic T lymphocyte response that inhibits tumor growth remarkably, prolongates survival, and yields a long-lasting immune memory response [125].

Platinum drugs have the same mechanism, that is, they all evolve DNA at the target site of action, and platinum atoms form cross-connections with DNA to antagonize its replication and transcription, so as to achieve anti-tumor effects. Based on inflammatory conditions induced by TLRa, cisplatin-induced immunosuppression-resistant DCs by the massive production of IL-10 with p38 MAPK and NF-κB signaling pathways activation, thereby distorting the differentiation of Th cells into Th2 and Tr1 cells, which might provide an opportunity for cancer cells to evade the immune system [130]. Similarly, oxaliplatin inhibited the differentiation of MDSCs into M1-like macrophages, leading to chemical resistance. Interestingly, TLR7/8 agonists R848 reversed this miseducation effect of oxaliplatin on MDSCs and enhanced the cytotoxic effect on cancer cells [131]. In those receiving four cycles of chemotherapy for the management of newly diagnosed, treatment-naïve advanced (stage IIIB/IV) NSCLC patients, CADI-05, a potent TLR2 agonist, and cisplatin-paclitaxel exhibited improved ORR, progression-free survival (PFS), and overall survival (OS) [132]. Likewise, a novel TLR7 agonist SZU-101 increased the efficacy of DOX, which produced powerful cytokines and advanced CTL responses, leading to the eradication of local and distant tumors in T-cell lymphoma-bearing mice [133]. These studies culminated in the development of biodegradable PLGA-PEG nanoparticles, a delivery vehicles for local, slow, and sustained release of DOX, poly(I:C), R848, and CCL20 chemokine for the potent candidate strategy to treat solid tumors resistant to first-line therapies [134]. The application of the combination treatment decreases the dosage of administered chemotherapy so as to mitigate the side effects in patients and likely attain better curative effects [135]. More studies are devoted to understanding which chemotherapeutic agents best collaborate with TLRa-mediated immunotherapy, what the best delivery regimen might be, and more importantly, what the mechanisms of this synergy for future therapies are.

### 6.2. Combination of Immunotherapy and Radiotherapy

Radiotherapy remains the primary non-surgical treatment for most cancer patients worldwide [136]. The anti-cancer effects of radiotherapy are based primarily on the destruction of chemical bonds within lipid membranes, proteins, and most importantly, between the bases in DNA. In addition to killing cells directly, radiation can also induce ICD repeatedly to cause abscopal effects [137]. However, radiotherapy is mainly treating localized solid tumors. On the other hand, typically targeted therapies often block a single pathway, allowing tumors to develop resistance by switching to other oncogenic pathways [138]. Many studies have confirmed that TLRa is a fantastic adjuvant [136], with the mechanism of cross-linking with the biological effects of radiotherapy [138]. Combined with radiotherapy, TLRa has been demonstrated to have the capacity for positive immune effects and reduced immune tolerance [136]. The studies have revealed that the combination of PGN and ionizing radiation (IR) enhanced tumor suppression and reduced the intestinal damage caused by IR compared to radiation alone [139].

Poly-ICLC, TLR3 agonists, combined with local nonlethal radiotherapy and local regional therapy, was safe and tolerable in patients with hepatocellular cancer (HCC) [40]. Furthermore, Poly-ICLC (such as Hiltonol) in combination with dendritic cell vaccine and multi-point SABR (stereotactic ablation radiotherapy) was also safe and showed preliminary clinical efficacy [140]. TLR7 agonists R837 in combination with γ-ionizing radiation (IR) reduced tumor growth through autophagy, with elevated CD8^+^ T cells and decreased Tregs and MDSCs in tumor lesions [141]. These studies culminated in the development of CMP-001, a CpG-A oligodeoxynucleotide TLR9 agonist delivered in a virus-like particle (VLP). Adding CMP-001 after RT increased the proportion of CD4^+^ and CD8^+^ T cells to enhance adaptive immunity [142]. Overall, these findings suggested that TLRa could be characterized as a radiosensitizer and an immune booster for radiation therapy in tumor treatment. Hence, TLRs and their ligands provide novel strategies for radiation protection of normal tissues during cancer radiotherapy [143]. The combination treatment of TLRa with radiotherapy may be a promising novel modality for cancer treatment, generally improving the treatment efficacy and potentially increasing survival rates in patients with cancer [135].

### 6.3. Combination of Immune Adjuvant and Phototherapy

Photothermal therapy (PTT) takes advantage of the heat generated by light absorbents to ablate tumor cells while enhancing immunogenicity in the tumor milieu through immunogenic cell death. However, single-mode PTT has a lower capability to eliminate distal or metastatic tumors due to more vulnerable immune activation. In addition, PTT-induced immune responses are easily suppressed if tumor temperature is above 45 °C, possibly owing to heat-induced chemokine and cytokine damage in the vascular system and temperature-induced stromal and tumor cell stress [144]. Meanwhile, challenges in immunotherapy remain a significant barrier, thus a logical strategy is the combination with these two therapies. The combination of phototherapy and cancer immunotherapy has been proven to have synergistic effects, promoting cancer regression and even gaining immune memory (Table 3) [145]. Zhou et al. developed a novel thyroid cancer treatment strategy rationally based on targeted hyperthermia, Hsp70 inhibitor quercetin thermo-sensitization, and LPS-enhanced immunogenicity [146]. Further work on a vaccine-like function in the presence of R837-containing nanoparticles as adjuvants, PLGA-ICG-R837 nanoparticles could be used for near-infrared laser triggered photothermal ablation of primary tumors to produce TAAs. When combined with anti-cytotoxic T lymphocyte antigen-4 (CTLA4), the remaining tumor cells in mice were attacked, contributing to restraining metastasis. This strategy supplied a powerful immune memory effect, which could offer protection against tumor recurrence after initial tumor elimination [147]. Another in situ anti-tumor vaccination was a multifunctional nanocomplex based on calcium crosslinked polyaspartic acid conjugated to both TLR7/8 agonist IMDQ and the photosensitizer IR780. Owing to various carboxyl groups in PASP, it could be crosslinked with calcium ions to form a pH-responsive nanocomplex. The system displayed that the repolarization of macrophages in TDLNs occurred as well as the proliferation and activity of CD4^+^ and CD8^+^ T cells in tumor sites, thus remarkably preventing tumor proliferation and metastasis [148].

Lin and colleagues reported dual-functional PLGA-ICG-R848 NPs in combination with PTT and immunotherapy for prostate cancer (PCa), promoting BMDCs maturation with the considerably increased proportions of CD11c^+^CD86^+^ and CD11c^+^CD80^+^ cells [152]. For an alternative approach, Hao and colleagues designed a smart nanovehicle equipped with multifunctional navigation for the precise delivery of IMDQ and ICD amplifiers. Upon exposure to NIR laser irradiation, both phototherapy and OXA enlarged ICD to release TAAs and DAMPs. Furthermore, the laser-triggered disintegration of CuS led to the collapse of release polymer coupling TLR7/8 agonist modified with mannose that activated DCs, enhancing the infiltration of T lymphocytes against cancer through manifold pathways. Collectively, the outcomes could reduce tumor burden, exert a robust antitumor immune response, and generate long-term immune protection against tumor recurrence [149].

Beyond intratumoral administration, numerous TLR7/8 based immunoadjuvant and photothermal combination therapies have been developed to enable intravenous administration of TLR7/8 agonists. A significant stride toward the systemic administration of TLR7/8 agonists-containing nanoparticles combined with photothermal therapy was made by Li and colleagues. They designed a semiconducting polymer nanoadjuvant (SPN_II_R) with a photothermally triggered cargo release for second near-infrared (NIR-II) photothermal immunotherapy (Figure 4B). Upon NIR-II light irradiation, SPN_II_R generated heat not only to ablate tumors and induce ICD robustly, but also to melt the lipid layer to release R848 on demand, which could enhance DCs maturation, thereby restraining the growth of primary and distant tumors and powerfully eliminating lung metastasis in a murine tumor model [106]. Notably, a combination of ICB was not addressed in this initial study; in a follow-up study, Cheng and coworkers took advantage of the ordered large-pore structure and easily chemically modified property of dendritic large-pore mesoporous silica nanoparticles (DLMSNs) for suppressing metastatic TNBC by combining photothermal ablation and immune remodeling. The tumor antigen is produced and released gradually through photothermal action and combined with AUNP-12 (anti-PD-1 peptide), isolated from AM@DLMSN@CuS/R848 in the weakly acidic tumor microenvironment, the synergistically elicited tumor vaccination, and T lymphocyte activation ability of immune remodeling to prevent TNBC rechallenge and metastasis (Figure 4C) [108].

Another recent attempt at the intravenous administration of TLR7/8 agonists was reported by Zhang and coworkers; they developed thermosensitive liposomes (TSLs) as delivery carriers that could enhance the infiltration and accumulation of CD8^+^ T cells. The intravenous administration of R848-TSLs, combined with local hyperthermia and αPD-1, increased the median survival by approximately three fold in comparison with non-treatment control or αPD-1 treatment only. All tumor-free mice treated locally with R848-TSLs in combination with αPD-1 developed a specific immunity to NDL cells and did not grow tumors after NDL tumors were rechallenged [150]. Another immune-adjuvant nanomedicine carrier was designed on the basis of polydopamine (PDA), loaded with TLR7/8 agonist R848 and carbon dots (CDs). The multitasking PDA-PEG-R848-CD NPs were able destroy 4T1 breast tumors under near-infrared laser irradiation as well as yield tumor-associated antigens, significantly potentiating the systemic efficacy of PD-L1 checkpoint blocking therapy by activating the innate and adaptive immune systems in vivo [151].

Moving beyond ICB combination treatment, Jia et al. developed an injectable local therapeutic platform based on NIR stimulating the drug release of thermal-sensitive PDLLA-PEG-PDLLA (PLEL) hydrogel to achieve synergistic photothermal immunotherapy for the prevention of postoperative recurrence of breast cancer (Figure 4D). Self-assembled multifunctional nanoparticles (RIC NPs) were composed of three therapeutic components, including indocyanine green (ICG) (a photothermal agent), R848, and CpG, ODNs. RIC NPs@PLEL ablated the residual tumor tissue and produced TAAs that served as in situ cancer vaccines for postoperative immunotherapy by inducing an intensive and persistent antitumor immune effect [109]. Hence, based on accumulating pieces of evidence, the combination of nanoparticles-based immune adjuvant and photothermal agents can be considered an exceptional access for activating the immune system to prevent recurrences and metastases [8].

### 6.4. Combination of Immune Adjuvants with Immune Checkpoint Inhibitors

Immune checkpoint inhibitors (ICIs) are one of the breakthrough treatments for many cancers [153]. ICIs targeting CTLA-4, PD-L1, and its receptor, PD-1, have shown significant antitumor activity and have been approved by regulators for multifarious adult cancers [154]. Anti-CTLA-4 antibodies inhibiting the effect of CTLA-4, so that CD28 binds to CD80/CD86 to motivate T cells [155], is achieved by spatially blocking the interaction of CTLA-4 with APCs or CD80 and CD86 on T cells [156], while anti-PD-1 antibodies target surface receptors expressed on activated T cells [156]. Nevertheless, only 10% to 30% of patients benefit from this immune checkpoint blocking, possibly due to the insufficient activation or inactivation of tumor-reactive CTLs or the inability to infiltrate the tumor, while anti-CTLA-4 in conjunction with anti-PD-1 antibodies can lead to severe immune-related toxicity in tumors treatment [157]. Therefore, some studies have adopted the method of binding immunoregulatory antibodies to innate immune response activators, such as TLRa, which have shown remarkable efficacy in different tumor models. Upon intratumoral injection, TLRa improved the efficacy of over-metastatic T cells and enhanced immunoregulatory antibodies, such as anti-CTLA-4 and anti-PD-1 [158].

For these reasons, diprovocim, a TLR1/2 agonist with superior adjuvanticity, is synergized with OVA and anti-PD-L1 therapy to inhibit the growth of melanoma tumors, generating long-term anti-tumor memory and prolonged survival [159]. Similarly, anti-PD-L1 antibodies in combination with ARNAX, a TLR3 specific agonist and OVA, brought complete remission to another PD-L1-high subline of EG7 [160]. Other ICI combinations have been reported, including one developed by Jeong and colleagues that was based on a novel TLR4 agonist, aqueous-formulated *E. coli*-derived MPLA (EcML-AF), with robust adjuvanticity that boosted the efficacy of anti-PD-1 treatment with elevated motivated tumor-specific CTLs in the B16F10-OVA models [161].

The combination of TLR9 agonists with the PD-1/PD-L1 blockade has shown an elevated number of multifunctional tumor antigen-specific T cells expressing TNF and IFNγ, and a proportion of memory precursor effector cells [162]. The anti-PD-1 associated with TLR9 agonists increased the ratio of CD8^+^ T cells to the suppressive myeloid stroma, while the anti-CTLA-4 combination increased the ratio of CD8^+^ T cells to Tregs in tumors [163]. Furthermore, the addition of TLR7 agonists progressed the proportion of M1 to M2 tumor-associated macrophages (TAMs) and promoted the infiltration of tumor-specific IFNγ -producing CD8^+^ T cells [164]. Another recent attempt at PD-1 blockade was demonstrated by Zhu and colleagues, wherein the glioma vaccine called STDENVANT consisted of glioma stem-like cell (GSC) lysate, DCs, and TLR9 agonists, CpG ODNs. A combination of anti-PD-L1 antibodies with STDENVANT offered a more prominent survival rate and reduced the number of Tregs in the brain [165].

### 6.5. Clinical Trials of TLRa Combination Therapy

A phase I/II trial evaluated the safety, tolerability, and initial efficacy of intratumoral injection of G100, a TLR4 agonist, after topical low-dose radiotherapy in patients with follicular lymphoma (NCT02501473). No new or unexpected toxicity was reported with pembrolizumab when used in combination with G100. Although all efficacy analyses were exploratory, low-dose radiation G100 injections displayed signs of clinical activity in FL patients. These data provided evidence of the safety of intertumoral injection and the administration of TLR4 agonists, as well as evidence of local and systemic antitumor immune responses [166]. Imiquimod, regarded as a TLR-7 agonist in combination with laser therapy, indocyanine green, was an available treatment in treating patients with stage III or stage IV melanoma (NCT00453050). This study had completed a phase I clinical trial and showed that in situ photoimmunotherapy (ISPI) using imiquimod was safe and well tolerated and easily applied in an outpatient setting [167]. Furthermore, a phase I/II study of imiquimod in combination with cyclophosphamide and radiotherapy in breast cancer patients with chest wall recurrence or cutaneous metastasis demonstrated that the addition of the immunomodulator cyclophosphamide (CTX) increased the antitumor response (NCT01421017). Nevertheless, the majority of clinical trials were faced with failure, as they were unable to achieve the desired therapeutic effect and patients were intolerant to adverse reactions. For example, this phase 1 study, the first in humans, evaluated TLR7/8 agonist MEDI9197 in combination with the PD-L1 inhibitor durvalumab and/or palliative radiotherapy (RT) in the treatment of advanced solid tumors (NCT02556463). Although MEDI9197 increased tumor CD8^+^ and PD-L1^+^ cells and induced type 1 and type 2 interferon and Th1 responses, MEDI9197-associated adverse events (AE) frequently occurred, resulting in dose-limiting toxicity (DLT) with cytokine release syndrome and, even more serious, in one patient (grade 5) resulted in two cycles of MEDI919 hemorrhagic shock associated with DLT after the rupturing of liver metastasis. Although no tumor response was observed in this clinical trial, MEDI9197 induced local and systemic immune activation, indicating the potential value in combination with other drugs [168]. Similarly, a phase I/II study of the intratumoral injection of the small Molecule TLR8 agonist VTX-2337 in combination with local radiation in low-grade B-cell lymphomas was terminated (NCT01289210). Gastrointestinal disorders such as nausea, injection site reactions, flu-like symptoms, and headache associated with nervous system disorders were reported in this study. Therefore, based on the above failures in clinical trials, effective therapeutic strategies are required that minimize adverse reactions to facilitate the successful clinical implementation of immunoadjuvant-based systems in clinical practice. This further confirms the significance of nanocarriers in TLRa delivery, as there are few TLRa nanocarriers currently being studied in clinical trials. In the design process of nanocarriers in the future, we should pay more attention to the biocompatibility and safety of nanomaterials, as well as the repeatability to facilitate large-scale production and clinical translation.

## 7. Conclusions and Future Perspective

The goal of cancer immunotherapy is to motivate T cells to recognize and dismantle tumor cells in an antigen-specific manner, which can be achieved by spurred APCs, primarily DCs and macrophages. TLR agonists stimulate DCs, generating the increased expression of costimulatory molecules and the secretion of proinflammatory cytokines, which in turn gives rise to T cell dilatation. TLR ligands are expressed not only in immune cells but partly in tumors, contributing to promoting immune escape or tumor growth. Hence, we should pay more attention to the selection of TLRa and cancer models in tumor therapy. Moreover, is the combination of multiple agonists necessarily and widely synergistic? The concentration on in-depth studies of the mechanism and assistance of appropriate biomarkers grant a window of opportunity to evaluate the ideal dosing framework to amplify the efficacy of immunotherapy in the future. Nevertheless, DC impetus results in proinflammatory production and even toxicity (also known as cytokine storms), which is a major bottleneck in the clinical transformation of TLR agonists. On the other hand, mice do not exhibit clinical symptoms of cytokine storms, owing to fundamental differences in the innate immunity between mice and humans and, therefore, may be of low predictive value for the human disease. It is another drawback that our mouse model deters investigation studies of cytokine storm-induced toxicity.

To date, accumulating evidence that both nano- and microparticle systems are capable of lessening the dose of immune adjuvants and supplying rising anti-tumor immunity in the targeted delivery of TLR agonists and tumor antigens compared to free drugs. In addition, the co-encapsulation of the immune adjuvant TLRa and OVA antigens into nanoscale and microparticle systems resulted in a durable anticancer immune response. Yet there is still the problem of synthesizing novel nanosystems that are adopted for cancer treatment, thereby restricting the clinical development of nanomedicines. For one, nondegradable byproducts from specifically designed drug delivery systems can be prone to long-term accumulation. It is essential to alter the unwanted pharmacokinetic profile of TLRa using a facile, yet efficient, strategy that holds high potential for clinical translation. The key aspects of the drug qualification, such as the persistence and stability of nanoparticles, need to be considered in the preparation process of nanoparticles. Drug screening, drug combination, dose, timing, and the mode of administration all impact the efficacy of the agent, thus the future research directions of TLRa with rational utilization are vital in order to realize the promising clinical value and make a leap into the future.

## Data Availability

Not applicable.

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
