# Peer review of "The Role of Toll-like Receptor Agonists and Their Nanomedicines for Tumor Immunotherapy"

_pharmaceutics, 2022, doi:10.3390/pharmaceutics14061228_

Round 1
Reviewer 1 Report
The authors nicely summarized the current state of using TLR agonists in anticancer immunotherapy. They have provided sufficient background and citations for the complex intricate pathways of immunomodulation. They have also summarized state-of-the-art strategies to advance cancer immunotherapy using TLRa-nanomedicine. The article is well-organized and well-timed. I recommend publishing this work with the following minor modifications:
The author should also include a separate table or section for clinical trials. It would be of great interest to the audience to get comments from experts in the field, particularly on those trials that have failed or terminated for scientific reasons.
Author Response
Comments: The author should also include a separate table or section for clinical trials. It would be of great interest to the audience to get comments from experts in the field, particularly on those trials that have failed or terminated for scientific reasons.
Reply: Thank you for your kind comments. We agree with the reviewer's point of view and we have included examples of TLR agonists currently being used in clinical trials, and specifically analyzed the reasons for termination of some clinical trials. The content has been supplemented in the last section of the “TLRa-based combination therapy” part as your suggestion and marked in yellow in the revised manuscript.

Reviewer 2 Report
The manuscript has scientific merit for describing the action mechanism of agonists of TLRs and the emerging applications of TLRa-loaded nanomedicines as state-of-the-art strategies to advance cancer immunotherapy. However, some points need to be reviewed before final acceptance.
Minor review
DCs and macrophages have a crucial role in the immune response via the NFKb and STAT3 axis in TME. In a paragraph and an image, the authors should describe the action mechanism of these APCs in a immune loop of modulation via TLRs. Both DCs and macrophages are cited in the whole text but the information is not concise. It seems a little confusing to those who read.
Author Response
Comments: DCs and macrophages have a crucial role in the immune response via the NFKb and STAT3 axis in TME. In a paragraph and an image, the authors should describe the action mechanism of these APCs in a immune loop of modulation via TLRs. Both DCs and macrophages are cited in the whole text but the information is not concise. It seems a little confusing to those who read.
Reply: Thank you for your kind comments. The literature related to the role of DCs and macrophages in TLRs regulation of immune circuitry has been supplemented in the second paragraph of the “Introduction” part as your suggestion, the revised details and supplied image were marked in yellow in the revised manuscript.
Reviewer 3 Report
Lingling Huang and colleagues present a quality and well-written review manuscript describing the role of Toll-like receptor agonists and their nanomedicines for tumor immunotherapy.
Authors first briefly depicted the patterns of TLRa, followed by the mechanism of agonists at those targets. Then they summarized the emerging applications of TLRa-loaded nanomedicines as state-of-the-art strategies to improve cancer immunotherapy. Finally, they outlined perspectives related to the development of nanomedicine-based TLRa combined with other therapeutic modalities for cancer immunotherapy.
Authors suggest that Toll-like receptor agonist (TLRa) as a candidate vaccine adjuvant has become one of the recent research hotspots in the cancer immunomodulatory field. However, numerous current systemic delivery of TLRa is inappropriate for clinical adoption due to the low efficiency and systemic adverse reactions. TLRa-loaded nanoparticles are capable of ameliorating the risk of immune-related toxicity and strengthening tumor suppression and eradication. Authors first briefly depicted the patterns of TLRa, followed by the mechanism of agonists at those targets. Then they summarized the emerging applications of TLRa-loaded nanomedicines as state-of-the-art strategies to advance cancer immunotherapy.
Authors cover TLRa in tumor therapy, combination of multiple TLRa for tumor therapy, application of multiple TLRa nanoparticles in tumor therapy, TLRa-based combination therapy and other topics.
Finally, authors conclude that it is essential to alter the unwanted pharmacokinetic profile of TLRa using a facile, yet efficient, strategy that holds high potential for clinical translation. The key aspects of the drug qualification, such as the persistence and stability of nanoparticles, need to be considered in the preparation process of nanoparticles. Drug screening, drug combination, dose, timing, and mode of administration all impact the efficacy of the agent, thus the future research directions of TLRa with rational utilization are vital in order to realize the promising clinical value and make a leap into the future.
===========
Other comments:
1) Please check for typos throughout the manuscript.
2) Authors are kindly encouraged to cite the following article that describes certain aspects of targeting tumor with immunotherapies (both innate and adaptive), including Toll-like receptor agonists.
DOI: 10.3389/fimmu.2021.707734.
Overall, the manuscript is highly valuable for the scientific community and should be accepted for publication.
Author Response
Comment 1: Please check for typos throughout the manuscript.
Reply: Thank you for your kind comments. We have checked for typos throughout the revised manuscript.
Comment 2: Authors are kindly encouraged to cite the following article that describes certain aspects of targeting tumor with immunotherapies (both innate and adaptive), including Toll-like receptor agonists.
Reply: Thank you for your kind comments. The literature related to the mechanisms of action by which TLRa targets tumors through innate and adaptive immunotherapy have been added in the third paragraph of the “Introduction” part as your suggestion, the revised details were marked in yellow in the revised manuscript.